# Corelease of acetylcholine and GABA from cholinergic forebrain neurons

Arpiar Saunders[†], Adam J Granger[†], Bernardo L Sabatini*

Department of Neurobiology, Howard Hughes Medical Institute, Harvard Medical School, Boston, United States

**Abstract** Neurotransmitter corelease is emerging as a common theme of central neuro-modulatory systems. Though corelease of glutamate or GABA with acetylcholine has been reported within the cholinergic system, the full extent is unknown. To explore synaptic signaling of cholinergic forebrain neurons, we activated *choline acetyltransferase* expressing neurons using channelrho-dopsin while recording post-synaptic currents (PSCs) in layer 1 interneurons. Surprisingly, we observed PSCs mediated by $GABA_A$ receptors in addition to nicotinic acetylcholine receptors. Based on PSC latency and pharmacological sensitivity, our results suggest monosynaptic release of both GABA and ACh. Anatomical analysis showed that forebrain cholinergic neurons express the GABA synthetic enzyme *Gad2* and the vesicular GABA transporter (*Slc32a1*). We confirmed the direct release of GABA by knocking out *Slc32a1* from cholinergic neurons. Our results identify GABA as an overlooked fast neurotransmitter utilized throughout the forebrain cholinergic system. GABA/ACh corelease may have major implications for modulation of cortical function by cholinergic neurons.

*For correspondence: bernardo_sabatini@hms.harvard.edu

[†]These authors contributed equally to this work

Competing interests: The authors declare that no competing interests exist.

## Introduction

For many years, neurons were thought to release only a single fast neurotransmitter (*Strata and Harvey, 1999*). This assumption led to classifying neuronal subtypes based on released neurotransmitter, a convention which helped predict a neuron's circuit function. However, many neuronal subtypes that release multiple fast neurotransmitters have now been described (*Hnasko and Edwards, 2012*). In some cases, the coreleased neurotransmitters have similar post-synaptic effects, such as inhibition mediated by GABA and glycine from spinal interneurons (*Jonas et al., 1998*). In other instances, the effects of the two neurotransmitters may be different and synergistic. For example, coreleased GABA and glutamate are thought to control the balance of excitation and inhibition in the lateral habenula (*Root et al., 2014*; *Shabel et al., 2014*). In neuromodulatory systems, synaptic release of fast neurotransmitters along with slow neuromodulators has emerged as a common theme. In addition to the impact of dopamine, stimulation of dopaminergic terminals from the ventral tegmental area and substantia nigra compacta activates glutamate-mediated excitatory currents in the nucleus accumbens (*Stuber et al., 2010*; *Tecuapetla et al., 2010*) and GABA-mediated inhibitory currents in the striatum (*Tritsch et al., 2012*, *2014*). Likewise, serotonergic neurons of the dorsal raphe can trigger glutamate-mediated currents in post-synaptic neurons of the ventral tegmentum and nucleus accumbens which contributes to the signaling of reward (*Liu et al., 2014*).

Several cholinergic neuron populations also release multiple neurotransmitters. Retinal starburst amacrine cells (SACs) differentially release GABA and acetylcholine (ACh) based on the pattern of light stimulation (*Lee et al., 2010*). In the central brain, selective activation of striatal cholinergic interneurons results in cholinergic and glutamatergic responses (*Gras et al., 2008*; *Higley et al., 2011*; *Nelson et al., 2014*). Similarly, the cholinergic projection from habenula excites interpedun-cular neurons through glutamate and ACh (*Ren et al., 2011*). Some evidence suggests that the basal forebrain cholinergic (BFC) system, which provides the major source of ACh to cortex, may corelease

**eLife digest** Neurons communicate with one another at junctions called synapses. When an electrical signal arrives at the presynaptic cell, it triggers the release of molecules called neurotransmitters into the synapse. These molecules then bind to receptor proteins on the postsynaptic cell, starting a chain of events that leads to the regeneration of the electrical signal in the second cell.

Broadly speaking, neurotransmitters are either excitatory, which means that they increase the electrical activity of the postsynaptic neurons, or they are inhibitory, meaning that they reduce postsynaptic activity. Initially, it was thought that neurons release only one type of neurotransmitter, but it is now known that this is not always the case. Many neurons within the spinal cord, for example, release two different inhibitory neurotransmitters, GABA and glycine, while some neurons in the midbrain release GABA and an excitatory neurotransmitter called glutamate.

Saunders, Granger, and Sabatini now provide the first direct evidence that cholinergic neurons in different regions of the forebrain also release two neurotransmitters. Collectively known as the 'forebrain cholinergic system', these cells are best known for producing the excitatory transmitter acetylcholine. However, Saunders et al. now show that this system also produces an enzyme that manufactures GABA, as well as a protein that pumps GABA into structures called vesicles, which are then released into the synapse.

Although this is not concrete evidence for the release of GABA, Saunders et al. also show—with a technique called optogenetics, which involves the use of light to control neuronal activity—that some of the neurons in this system can trigger inhibitory responses in postsynaptic cells. Moreover, these responses can be blocked using drugs that occupy GABA receptors, or by using genetic techniques to delete the GABA-pumping protein from cholinergic neurons.

Taken together, the results of these experiments strongly suggest that the cholinergic neurons throughout the forebrain—unlike, for example, the cholinergic neurons in the midbrain, the region of the brain that controls movement—possess the molecular machinery needed to produce and release GABA, in addition to acetylcholine. Given that the cholinergic system has a key role in cognition and is particularly susceptible to degeneration in Alzheimer's disease, the ability of these neurons to release GABA release could have widespread implications for the study and understanding of brain function.

GABA. The dorsal-most BFC neurons, which belong to the globus pallidus externus (GP), express molecular markers for GABA synthesis and vesicular packaging and trigger $GABA_A$ receptor currents in GP and cortex when activated (*Tkatch et al., 1998*; *Saunders et al., 2015*). We therefore asked whether GABA corelease was a general feature of forebrain cholinergic neurons.

To address this question, we selectively activated cholinergic fibers in the cortex, with the goal of identifying synaptic events triggered by endogenous release from forebrain cholinergic neurons. Recording from layer 1 interneurons, we observed not only the expected excitatory post-synaptic currents (EPSCs) mediated by nicotinic ACh receptors (nAChRs), but an unexpected inhibitory post-synaptic current (IPSC) mediated by $GABA_A$ receptors. IPSCs insensitive to nAChR antagonists had onset latencies slightly faster than the nicotinic EPSCs (nEPSCs), and could be directly evoked under pharmacological conditions in which action potentials were blocked, suggesting cholinergic neurons were directly releasing GABA in addition to ACh. Indeed, we found that cholinergic neurons throughout the forebrain commonly coexpressed the GABA synthetic enzyme GAD65 (*Gad2*), and the vesicular GABA transporter (*Slc32a1*), indicating that these neurons possess the necessary cell machinery for GABA transmission. Finally, we show that conditional deletion of *Slc32a1* selectively in cholinergic neurons eliminates monosynaptic IPSCs while leaving nEPSCs intact, confirming the direct release of GABA from cholinergic terminals. These experiments suggest a previously overlooked capability of the cholinergic system to use GABA in synaptic signaling.

## Results

To explore the effects that cholinergic neurons have on cortical circuitry, we used double transgenic mice to optogenetically activate neurons that express endogenous *choline acetyltransferase* (*Chat*).

These mice carried a knock-in allele linking Cre recombinase expression to the *Chat* locus through an internal-ribosome entry site (*Chat* [i-Cre]) as well as a Cre-activated channelrhodopsin-EYFP fusion allele (*Rosa26* [lsl-ChR2-EYFP]). *Chat* [i-Cre]; *Rosa26* [lsl-ChR2-EYFP] mice expressed ChR2-EYFP throughout the forebrain, recapitulating known patterns of *Chat* expression in cortex (Ctx), striatum, globus pallidus externus (GP), and nucleus basalis (NB, *Figure 1A*). To test whether ChR2[+] cells express endogenous *Chat*, we performed ChAT immunohistochemistry on sections of *Chat* [i-Cre]; *Rosa26* [lsl-ChR2-EYFP] mice (*Figure 1B*). We focused on those *Chat*[+] forebrain neurons positioned to innervate the cortex, including local *Chat*[+] interneurons and the subcortical projections arising from the GP/NB. In both regions, ChR2[+] neurons were immunopositive for ChAT, confirming our ability to selectively activate endogenous *Chat*[+] inputs to cortex.

To identify the synaptic signaling mechanisms engaged by activation of the cortical cholinergic system, we targeted layer 1 interneurons for whole-cell voltage-clamp recordings in acute brain slices. Layer 1 is strongly innervated by ChAT[+] cells of the basal forebrain across species (*Mesulam, 1995*; *Mechawar et al., 2000*) including in *Chat* [i-Cre]; *Rosa26* [lsl-ChR2-EYFP] mice, where ChR2-EYFP is expressed in a dense plexus (*Figure 1C,D*). As expected, in a subset of interneurons (n = 41 of 58 cells from 9 mice), we observed robust excitatory postsynaptic currents (EPSCs) at −70 mV in response to brief pulses of blue light (2–7 ms, *Figure 1E*, left). These EPSCs were not blocked by NBQX and CPP, ruling out a glutamatergic source, but were blocked by the nicotinic ACh receptor (nAChR) antagonists DHβE, MLA, and MEC, confirming their cholinergic identity (nEPSCs). nEPSCs displayed a typical biphasic response, with a fast component, likely mediated by synaptic receptors containing the low-affinity α7 nAChR subunit, and a slow component, likely mediated by non-synaptic receptors expressing the high-affinity non-α7 subunits (*Bennett et al., 2012*).

In addition to the expected nEPSCs recorded at −70 mV, we also observed barrages of outward inhibitory postsynaptic currents (IPSCs) at 0 mV, indicative of signaling through GABA receptors (n = 28 of 58 cells, *Figure 1E*, center). One possible explanation for these IPSCs could be ACh-mediated feed-forward activation of local interneurons, resulting in disynaptic release of GABA. Indeed, when nAChR antagonists were applied, the delayed outward IPSCs disappeared. However, in a subset of recorded cells IPSCs remained (n = 9 of 58 cells, *Figure 1F*, right), suggesting that these PSCs were not dependent on nAChR signaling.

To test if nAChR-resistant IPSCs are caused by direct release of GABA from cholinergic fibers, we bath applied the voltage-gated sodium channel antagonist TTX, which blocked light-evoked IPSCs (*Figure 1F,G*). In the presence of TTX, IPSCs could be rescued by enhancing ChR2-mediated depolarization with the voltage-gated potassium channel blocker 4AP. Rescued IPSCs were subsequently blocked by the GABA$_A$ receptor antagonist SR95531 (n = 5 cells from 4 mice). Moreover, nAChR-resistant 'direct' IPSCs had faster average onsets than both nEPSCs and nAChR-sensitive 'indirect' IPSCs (nEPSCs, 4.0 ± 0.2 ms, n = 41 cells; direct IPSCs, 2.5 ± 0.2, n = 9 cells; indirect IPSCs, 11.8 ± 2.3, n = 19 cells from 9 mice, *Figure 1H*). These data suggest direct IPSCs are independent of nAChR signaling and mediated by GABA$_A$ receptors, consistent with monosynaptic release of GABA by cholinergic neurons. In support of this possibility, gene expression analyses have suggested some populations of *Chat*[+] subcortical neurons contain the synthetic machinery for GABA (*Kosaka et al., 1988*; *Tkatch et al., 1998*).

GABA corelease has been observed in other neuromodulatory systems, namely from dopaminergic neurons of the substantia nigra (*Tritsch et al., 2012*). In those neurons, GABA is co-packaged with dopamine into vesicles by the transporter VMAT2 (*Slc18a2*), instead of by the typical vesicular transporter for GABA (VGAT, encoded by *Slc32a1*), which is necessary for packaging in most GABAergic neurons (*Wojcik et al., 2006*; *Tong et al., 2008*; *Kozorovitskiy et al., 2012*). To assess whether cholinergic neurons could use VGAT to package GABA into vesicles, we tested for *Slc32a1*/ChAT co-expression throughout the brain, including in cortex, GP, NB, diagonal band of broca (DBB)/medial septum (MS), and pedunculopontine nucleus (PPN, *Figure 2A*, top). We labeled cells expressing endogenous *Slc32a1* using double-transgenic mice that link Cre recombinase expression to the *Slc32a1* locus (*Slc32a1*[i-Cre]) and carry a zsGreen Cre reporter allele (*Rosa26* [lsl-zsGreen]). Subsequent ChAT immunolabeling on sagittal and coronal sections from *Slc32a1*[i-Cre]; *Rosa26* [lsl-zsGreen] mice was used to examine coexpression. Since zsGreen accumulates in somata and does not diffuse throughout the cytoplasm, this strategy allowed the clear identification of ChAT[+] soma free from background fluorescence caused by Cre[+] axons and dendrites. In cortex, GP, NB, and MS/DBB, nearly all of ChAT[+] cell were *Slc32a1*[+] (zsGreen[+]/ChAT[+], from 3 mice: Ctx, 628/628; GP, 238/243; NB, 598/624; MS/DBB;

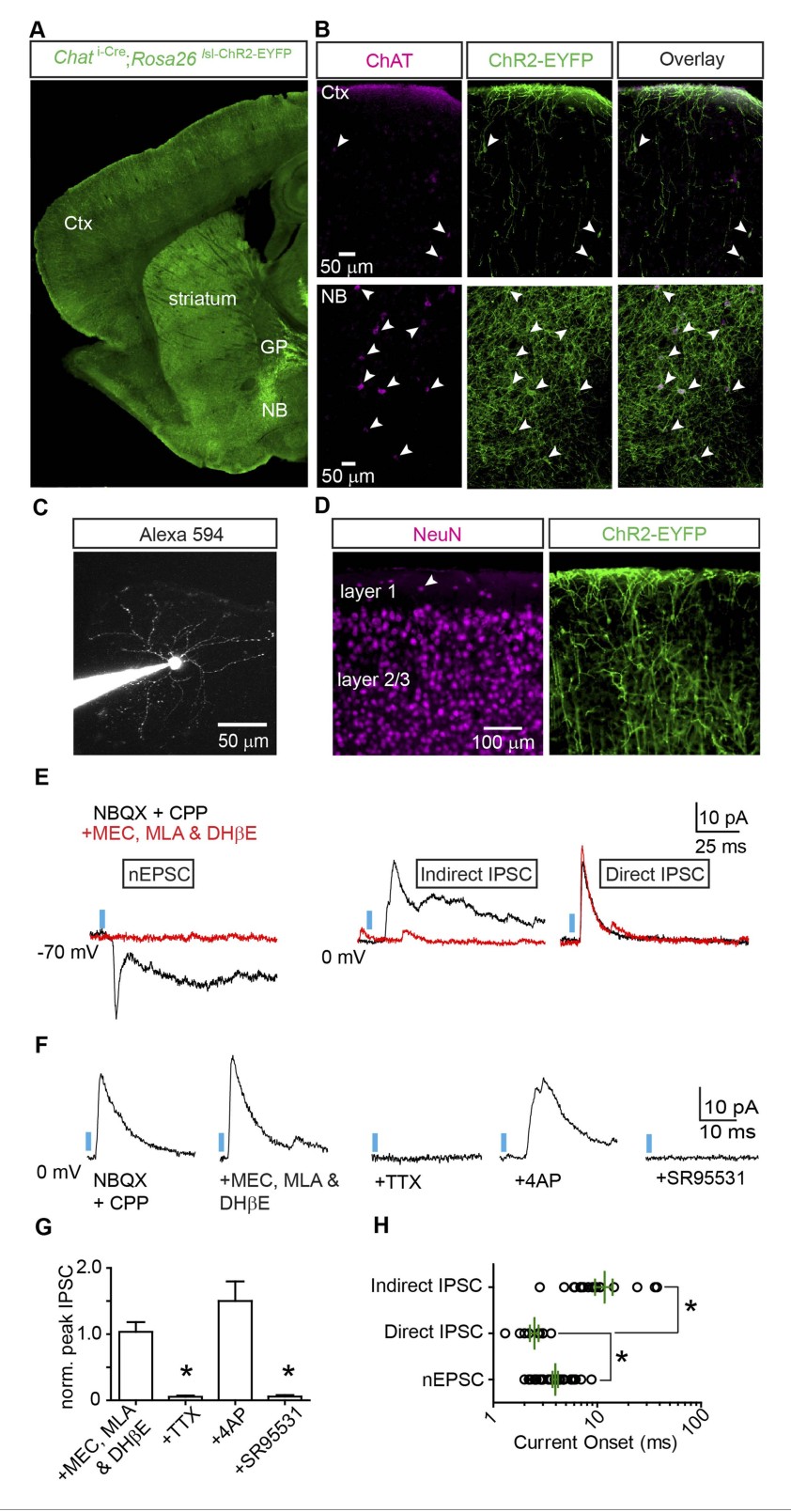

**Figure 1**. Optogenetic stimulation of cortical *Chat*+ fibers evokes fast, monosynaptic GABA_A receptor-mediated currents. (**A**) Low-magnification view of sagittal section from *Chat* [i-Cre]; *Rosa26* [lsl-ChR2-EYFP] mouse forebrain. ChR2-EYFP (green) is expressed in the nucleus basalis (NB), globus pallidus externus (GP), striatum and cortex (Ctx). (**B**) Higher-magnification view of ChR2-EYFP expression combined with ChAT immunostaining (magenta) in frontal

*Figure 1. continued on next page*

*Figure 1. Continued*

cortex (*top*) and nucleus basalis (*bottom*). Arrowheads indicate cells immunopositive for ChAT. (**C**) Example 2-photon stack from a layer 1 interneuron following whole-cell recording and dialysis with Alexa Fluor 594. (**D**) High-magnification view of ChR2-EYFP⁺ somata and fibers in layers 1 and 2/3 of frontal cortex. NeuN immunostain (magenta) highlights the distribution of neuronal somata across layers. Arrowhead indicates an example layer 1 interneuron surrounded by ChR2-EYFP fibers. (**E**) Example PSCs from voltage-clamp recordings of three different layer 1 interneurons in response to blue light stimulation (blue bar) of ChR2⁺ cholinergic fibers. Neurons were voltage-clamped at −70 mV (*left*) to isolate EPSCs or at 0 mV (*middle and right*) to isolate IPSCs. PSCs recorded in the presence of glutamate receptor antagonists CPP and NBQX are shown in black and after bath application of nAChR antagonists (MEC, MLA, and DHβE) in red. (**F**) Example light-evoked IPSCs from a layer 1 interneuron in a Chat ^i-Cre^; *Rosa26* ^lsl-ChR2-EYFP^ mouse voltage-clamped at 0 mV in the presence of CPP and NBQX (baseline) and following subsequent bath application of (from *left* to *right*) nAChR antagonists, TTX, 4AP, and SR95531. (**G**) Summary graph of IPSC peaks normalized to baseline (n = 5 cells from 4 mice). Asterisk, condition vs baseline p < 0.05, Mann–Whitney test). (**H**) Onset latencies for monosynaptic nEPSCs (n = 41 cells), monosynaptic IPSCs (n = 9 cells), and polysynaptic IPSCs (n = 19 cells from 9 mice). Mean (±sem) are shown in green. Asterisk, p < 0.05, Mann–Whitney test.

560/601, *Figure 2A,B*). In contrast, ChAT⁺ cells of the PPN very rarely expressed *Slc32a1* (3/157, *Figure 2B*). These data suggest that in both cortical and subcortical forebrain regions, ChAT⁺ cells express the canonical molecular machinery to package GABA into vesicles. However, since ChAT⁺ neurons in the brainstem PPN do not express *Slc32a1*, this GABAergic marker is not a ubiquitous feature of the central cholinergic system.

GABA is synthesized by one of two GABA synthetic enzymes, GAD65 or GAD67, encoded by the genes *Gad2* or *Gad1*, respectively. GAD67 is expressed largely in cell bodies and is thought to be responsible for synthesizing GABA for general metabolic cell functions, whereas GAD65 expression is more prominent in axon terminals and is thought to mediate the majority of synthesis of synaptic GABA (*Soghomonian and Martin, 1998*). We examined if cholinergic neurons expressed GAD67 by immunostaining for ChAT in brain sections from knock-in mice where GFP replaces the first exon of the *Gad1* gene (*Gad1^GFP^*). We detected only minor overlap between neurons that stained positive for ChAT and those that expressed GFP in the NB, GP, MS/DBB, or PPN, except for cortical ChAT⁺ interneurons, where we observed significant overlap (GFP+/ChAT + neurons: Ctx: 122/136; MS/DBB, 1/573; NB, 0/439; GP, 7/413; PPN, 12/246, *Figure 3—figure supplement 1*). This suggests that within the major subcortical cholinergic projections, GAD67-mediated GABA synthesis does not occur in cholinergic neurons. To test for coexpression of ChAT and GAD65, we immunostained sections of knock-in mice where Cre recombinase was targeted to the endogenous *Gad2* gene (*Gad2* ^i-Cre^) and visualized Cre expression with the zsGreen Cre reporter. In contrast to *Gad1*, in cortex, GP, NB, and MS/DBB of *Gad2* ^i-Cre^; *Rosa26* ^lsl-zsGreen^ mice, most ChAT⁺ neurons were *Gad2*⁺ (zsGreen⁺/ChAT⁺, from 4 mice: Ctx, 518/519; GP, 273/372; NB, 860/934; MS/DBB, 673/685, *Figure 3A,B*). In the brainstem, however, few of the ChAT⁺ cells were *Gad2*⁺ (PPN, 6/110, *Figure 3A,B*). This ChAT co-expression pattern for *Gad2* is similar to *Slc32a1*, suggesting that forebrain cholinergic neurons possess the necessary cellular machinery to both synthesize and package synaptic GABA.

Though the fast onset and persistence of GABAergic IPSCs in the presence of nAChR antagonists and TTX/4AP argues strongly for direct release of GABA from cholinergic terminals, we wished to confirm this finding using conditional genetics. To determine if GABA release is indeed monosynaptic, we took advantage of the observation that cholinergic neurons express *Slc32a1*. If GABA is released directly from cholinergic neurons, then conditional knock-out of *Slc32a1* selectively in these cells should abolish GABA release. We therefore bred triple transgenic mice which carried conditional (floxed) *Slc32a1* alleles in addition to *Chat* ^i-Cre^ and *Rosa26* ^lsl-ChR2-EYFP^. These mice allowed us to compare the optogenetic stimulation of wild-type cholinergic neurons (*Slc32a1*^+/+^) to those lacking *Slc32a1* (*Slc32a1*^fl/fl^) in acute brain slices. In voltage-clamp recordings from layer 1 interneurons, we observed fast onset IPSCs in 31% of cells recorded in *Chat* ^i-Cre^; *Slc32a1*^+/+^ mice (n = 5 of 16 from 2 mice), but no direct IPSCs in *Chat* ^i-Cre^; *Slc32a1*^fl/fl^ mice (n = 0 of 34 cells from 4 mice, *Figure 4A,B*). In contrast, the proportion and average peak amplitude of nAChR responses remains unchanged between *Slc32a1*^+/+^ and *Slc32a1*^fl/fl^ mice (*Figure 4A,B*). These data demonstrate that GABA but not

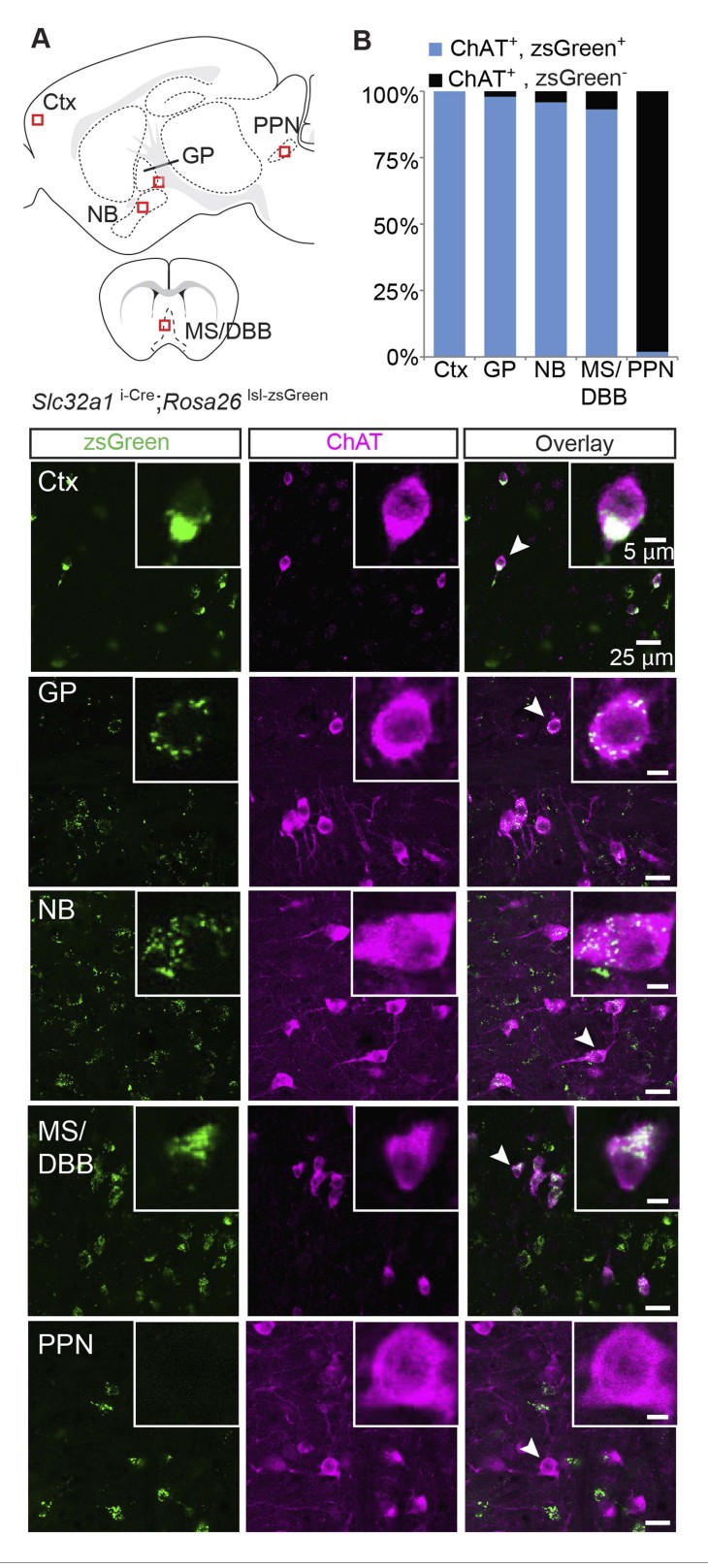

**Figure 2**. Immunopositive ChAT cells in the forebrain express *Slc32a1*. (**A**) *Top*, sagittal and coronal schematic views of a mouse brain showing cholinergic regions of interest. Red boxes indicate approximate locations for magnified regions below. *Bottom*, example single-plane image from a confocal stack from sections of a *Slc32a1*[i-Cre]; *Rosa26*[lsl-zsGreen] mouse immunostained for ChAT (magenta) and reporting Cre expression (green). Ctx, cortex; GP, globus pallidus externus; *Figure 2. continued on next page*

*Figure 2. Continued*

NB, nucleus basalis; MS/DBB, medial septum/diagonal band of broca; PPN, pedunculopontine nucleus. Insets show magnified view of individual neurons indicated by the white arrowhead. (**B**) Quantification of colocalization between cells expressing zsGreen Cre reporter and ChAT immunostain by brain region (zsGreen[+]/ChAT[+], from 3 mice: Ctx, 628/628; GP, 238/243; NB, 598/624; MS/DBB, 560/601; PPN, 3/157).

ACh release from cholinergic neurons relies on cell autonomous *Slc32a1*, ruling out a disynaptic mechanism.

## Discussion

Here, we provide evidence that the cortical cholinergic system is capable of GABAergic neurotransmission. In response to optogenetic stimulation of neurons expressing endogenous *Chat*, we observed PSCs mediated by both nAChRs and GABA$_A$ receptors in layer 1 interneurons. A subset of the evoked IPSCs appeared to be monosynaptic, based on latency and pharmacological analyses. In support of their GABAergic nature, the ChAT[+] neurons which innervate cortex—local interneurons and subcortical projections arising from the GP/NB—express *Gad2* and *Slc32a1*, the canonical molecular machinery for GABA synthesis and vesicular packaging. Indeed, conditional knock-out of *Slc32a1* selectively in cholinergic neurons eliminates light-evoked monosynaptic IPSCs. These genetic results confirm that cholinergic neurons release GABA directly. Cholinergic GABA release is likely to be a feature of most, but not all, central cholinergic neurons: the ChAT[+] neurons of the MS and DBB—which innervate the hippocampus—also express *Gad2* and *Slc32a1*, whereas those of the midbrain PPN do not.

The mode of neurotransmitter corelease can vary across neuron classes. In some instances, both neurotransmitters are released from the same synaptic vesicles. This is the case for GABA/glutamate corelease onto neurons of the lateral habenula, where individual miniature EPSCs can be observed with dual GABA/glutamate components (*Shabel et al., 2014*). Copackaging in individual vesicles is also the case when the same vesicular transporter loads both neurotransmitters. For example, GABA is packaged in dopaminergic neurons by VMAT2, which also packages dopamine (*Tritsch et al., 2012*). Similarly, both GABA and glycine packaging in spinal interneurons rely on VGAT (*Wojcik et al., 2006*). However, corelease from separate pools of synaptic vesicles also occurs. In retinal SACs, release of GABA and ACh can be functionally separated through patterned light stimulation or pharmacology. In *Chat*[+] GP axons in cortex, GABA and ACh appear to be released from distinct vesicular pools which can be located within the same or neighboring pre-synaptic terminals (*Saunders et al., 2015*). Though the precise mechanism by which each forebrain cholinergic population coreleases GABA and ACh remain unclear and further experiments are merited to explore how GABA/ACh cotransmission is regulated, the differences in the proportion of layer 1 interneurons showing ACh and GABA responses suggest some form of segregated release.

The cholinergic system's function in promoting attention, alertness, and learning has classically been attributed to acetylcholine and its action as a diffuse volume transmitter, affecting cortical activity at relatively slow time scales. This model is supported by anatomical evidence showing widespread distribution of cholinergic fibers through all cortical layers with significant separation between the sites of release and ACh receptors (*Descarries and Mechawar, 2000*; *Mechawar et al., 2000*). In addition, many in vitro pharmacological experiments have shown that ACh receptors can shape the signaling of other neurotransmitter systems, by altering properties of presynaptic release, synaptic plasticity, or the intrinsic excitability of targeted neurons (*Picciotto et al., 2012*). However, more recent work has focused on the participation of ACh in rapid, wired neurotransmission, acting at tightly apposed synapses (*Sarter et al., 2009*; *Poorthuis et al., 2014*; *Sarter et al., 2014*). Behaviorally relevant sensory cues can cause a fast, time-locked spike in ACh concentration, suggesting that ACh may mediate detection of that cue (*Parikh et al., 2007*). In addition, fast onset currents mediated by nAChRs can be recorded in cortical interneurons following optogenetic activation of cholinergic fibers (*Arroyo et al., 2012*; *Bennett et al., 2012*). Our finding that cholinergic neurons also elicit fast-onset synaptic GABA$_A$ responses lends further support to the notion that the cholinergic system can rapidly affect cortical computations by acting at classical synapses.

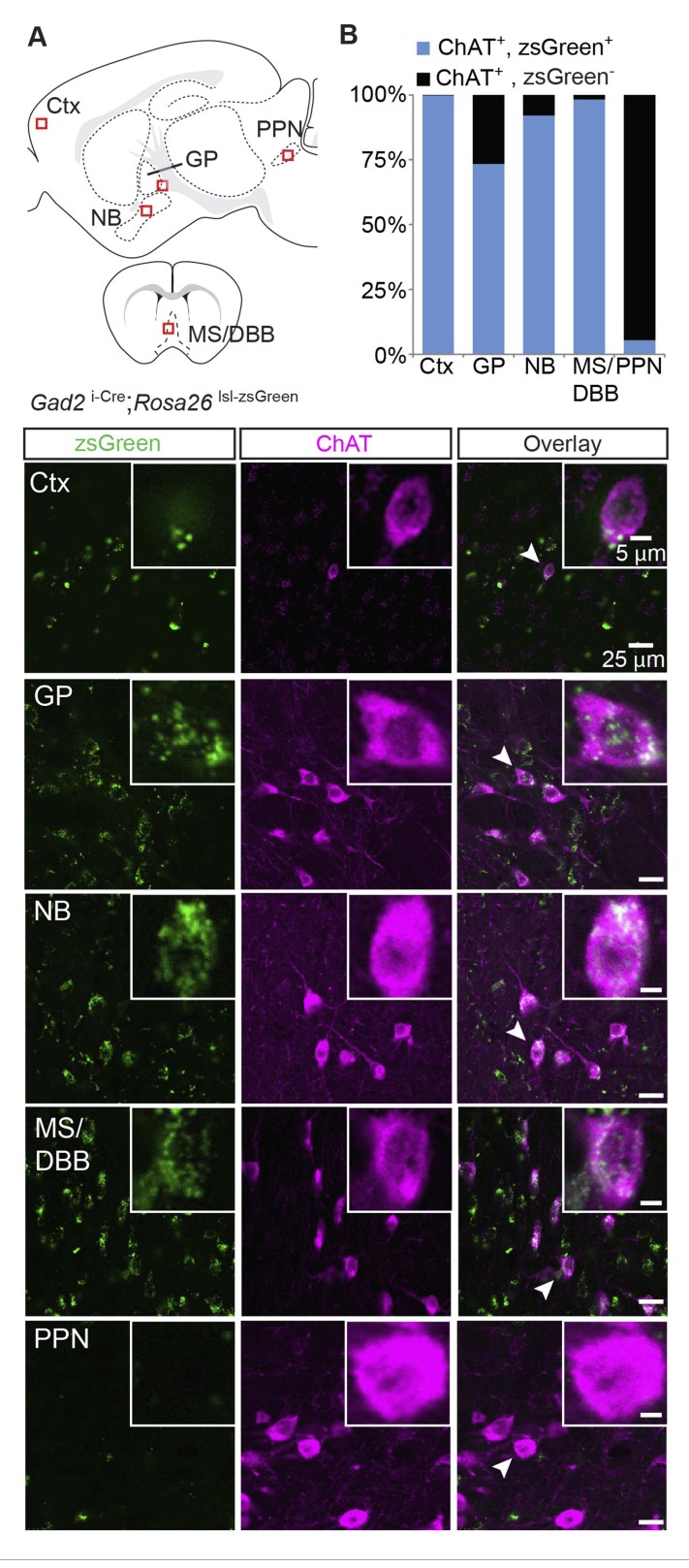

**Figure 3**. Immunopositive ChAT cells in the forebrain express *Gad2*. (**A**) *Top*, sagittal and coronal schematic views of a mouse brain showing cholinergic regions of interest. Red boxes indicate approximate locations for magnified regions below. *Bottom*, example single-plane image from a confocal stack from sections of a *Gad2* [i-Cre]; *Rosa26* [lsl-zsGreen] mouse immunostained for ChAT (magenta) and reporting Cre expression (green). *Figure 3. continued on next page*

*Figure 3. Continued*

Ctx, cortex; GP, globus pallidus externus; NB, nucleus basalis; MS/DBB, medial septum/diagonal band of broca; PPN, pedunculopontine nucleus. Insets show magnified view of individual neurons indicated by the white arrowhead. (**B**) Quantification of colocalization between cells expressing zsGreen Cre reporter and ChAT immunostain by brain region (zsGreen⁺/ChAT⁺, from 4 mice: Ctx, 518/519; GP, 273/372; NB, 860/934;MS/DBB, 673/685; PPN, 6/110).
The following figure supplement is available for figure 3:

**Figure supplement 1**. Immunopositive ChAT cells of the cortex but not major subcortical projections express *Gad1*.

---

GABA corelease from cholinergic forebrain neurons may affect cortical function in several ways. First, at the circuit level, GABA release could act in a manner that reinforces the emerging concept that the cholinergic system disinhibits cortical firing. ACh release from basal forebrain neurons excites layer 1 and VIP⁺ interneurons, which in turn inhibit local interneurons that target principle neurons of the cortex (*Letzkus et al., 2011*; *Pinto et al., 2013*; *Fu et al., 2014*). Depending on the timing and targeted cell types, GABA corelease could conceivably enhance this effect by inhibiting local interneurons, thereby promoting cortical activity. Second, nAChRs can regulate both pre and post-synaptic GABAergic signaling. For example, in hippocampal interneurons, post-synaptic nAChRs are present in inhibitory synapses (*Fabian-Fine et al., 2001*) and when activated, reduce GABA$_A$ receptor-mediated IPSCs in a Ca$^{2+}$ and PKC-dependent manner (*Wanaverbecq et al., 2007*; *Zhang and Berg, 2007*). Thus coreleased ACh and GABA could interact to modulate local synaptic signaling. Lastly, experiments poisoning or stimulating the basal forebrain cholinergic system have demonstrated that activity within this projection is necessary and sufficient for plasticity in sensory cortices (*Kilgard and Merzenich, 1998*; *Weinberger, 2004*; *Ramanathan et al., 2009*). While ACh alone can induce functional changes in cortical circuits (*Metherate and Weinberger, 1990*), GABA may also contribute to synaptic rewiring in vivo. Addressing these questions experimentally will benefit from future work to clarify the basic synaptic anatomy and biochemical regulation of ACh/GABA corelease. Given the presence of GABA signaling machinery throughout the distinct forebrain cholinergic systems, corelease likely has a significant and fundamental effect on brain activity and cognition.

## Materials and methods

### Mice

Cre recombinase was targeted to specific cell types using knock-in mice to drive Cre expression under endogenous gene-specific regulatory elements using an internal ribosome entry site. Cre knock-in mice for *choline acetyltransferase* (*Chat*) (*Rossi et al., 2011*) and vesicular GABA transporter (*Slc32a1*) (*Tong et al., 2008*) were provided by Brad Lowell (Beth Israel Deaconess Medical Center) and are available from the Jackson Labs (Bar Harbor, ME; *Chat* [i-Cre], stock #006410; *Slc32a1*[i-Cre], stock #016962). *Gad2* [i-Cre] mice were purchased from Jackson Labs (stock #010802) (*Taniguchi et al., 2011*). *Gad1*[GFP] knock-in mice replace elements of *Gad1* coding sequence with GFP (*Tamamaki et al., 2003*). We did not distinguish between mice hetero or homozygous for transgenic alleles except where indicated. All experimental manipulations were performed in accordance with a protocol (#03551) approved by the Harvard Standing Committee on Animal Care following guidelines described in the US National Institutes of Health Guide for the Care and Use of Laboratory Animals.

### Fixed tissue preparation, immunohistochemistry, and imaging

Mice aged post-natal day 25–124 were deeply anesthetized with isoflurane and transcardially perfused with 4% paraformaldehyde (PFA) in 0.1 M sodium phosphate buffer (1× PBS). Brains were post-fixed for 1–3 days, washed in 1× PBS, and sectioned (40–50 μm) coronally or sagittally using a Vibratome (Leica Microsystems, Buffalo Grove, IL). For ChAT or NeuN immunohistochemistry, slices were incubated in a 1× PBS blocking solution containing 5% normal horse serum and 0.3% Triton X-100 for 1 hr at room temperature. Slices were then incubated overnight at 4°C in the same solution containing anti-choline acetyltransferase antibody (AB144P; 1:100; Millipore, Billerica, MA) or NeuN

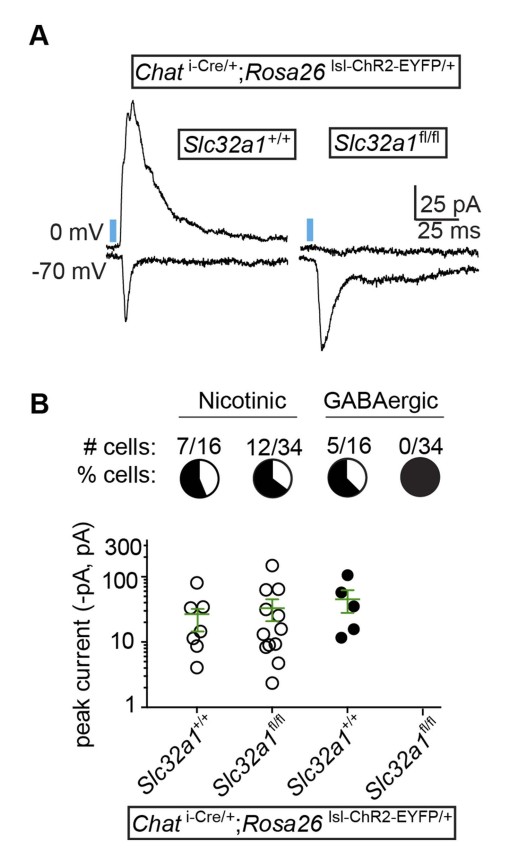

**Figure 4**. GABA release from cortical ChAT[+] axons requires *Slc32a1*. (**A**) Example light-evoked nEPSCs and IPSCs from four different layer 1 interneurons voltage-clamped at −70 or 0 mV from *Chat* [i-Cre]; *Rosa26* [lsl-ChR2-EYFP] mice with wild-type cholinergic neurons (*Slc32a1*[+/+]) or conditional *Slc32a1* knock-out (*Slc32a1*[fl/fl]). (**B**) *Top*, the number and proportion of layer 1 interneurons in which light-evoked nEPSCs or direct IPSCs were detected from *Chat* [i-Cre]; *Rosa26* [lsl-ChR2-EYFP] mice with wild-type *Slc32a1* alleles (*Slc32a1*[+/+], from 2 mice) or following conditional *Slc32a1* knock-out (*Slc32a1*[fl/fl], from 4 mice). *Bottom*, PSC peaks for each condition. Means (±sem) are shown in green.

(MAB377; 1:100; Millipore). The next morning, sections were washed three times for five minutes in 1× PBS and then incubated for 1 hr at room temperature in the blocking solution containing donkey anti-goat Alexa 647 or Alexa 594 (for ChAT) or anti-mouse Alexa 647 (for NeuN) (1:500; Molecular Probes, Eugene, OR). Slices were then mounted on slides (Super Frost). After drying, slices were coverslipped with ProLong antifade mounting media containing DAPI (Molecular Probes) and imaged with an Olympus VS110 slide scanning microscope using the 10× objective. Confocal images (1–2 μm optical sections) were acquired with an Olympus FV1000 laser scanning confocal microscope (Harvard Neurobiology Imaging Facility) through a 60× objective. Colabeling quantification was carried on images obtained from the Olympus VS110 slide scanning microscope using ImageJ.

## Slice preparation

Acute brain slices were obtained from mice aged post-natal day 30–128 using standard techniques. Mice were anesthetized by isoflurane inhalation and perfused through the heart with ice-cold artificial cerebrospinal fluid (ACSF) containing (in mM) 125 NaCl, 2.5 KCl, 25 NaHCO$_3$, 2 CaCl$_2$, 1 MgCl$_2$, 1.25 NaH$_2$PO$_4$, and 11 glucose (~308 mOsm·kg$^{-1}$). Cerebral hemispheres were removed, placed in ice-cold choline-based cutting solution (consisting of [in mM]: 110 choline chloride, 25 NaHCO$_3$, 2.5 KCl, 7 MgCl$_2$, 0.5 CaCl$_2$, 1.25 NaH$_2$PO$_4$, 25 glucose, 11.6 ascorbic acid, and 3.1 pyruvic acid), blocked, and transferred into a slicing chamber containing ice-cold choline-based cutting solution. Sagittal slices (300–350 μm thick) were cut with a Leica VT1000s vibratome and transferred to a holding chamber containing ACSF at 34°C for 30 min and then subsequently at room temperature. Both cutting solution and ACSF were constantly bubbled with 95% O$_2$/5% CO$_2$.

## Acute slice electrophysiology and two-photon imaging

Individual slices were transferred to a recording chamber mounted on a custom built two-photon laser scanning microscope (Olympus BX51WI) equipped for whole-cell patch-clamp recordings and optogenetic stimulation. Slices were continuously superfused (3.5–4.5 ml·min$^{-1}$) with ACSF warmed to 32–34°C through a feedback-controlled heater (TC-324B; Warner Instruments). Cells were visualized through a water-immersion 60× objective using differential interference contrast (DIC) illumination. Epifluorescence illumination was used to identify those layer 1 interneurons surrounded by ChR2-EYFP processes. Patch pipettes (2–4 MΩ) pulled from borosilicate glass (G150F-3; Warner Instruments) were filled with a Cs$^+$-based low Cl$^-$ internal solution containing (in mM) 135 CsMeSO$_3$, 10 HEPES, 1 EGTA, 3.3 QX-314 (Cl$^-$ salt), 4 Mg-ATP, 0.3 Na-GTP, 8 Na$_2$-Phosphocreatine (pH 7.3 adjusted with CsOH; 295 mOsm·kg$^{-1}$) for voltage-clamp recordings. Series resistance (<25 MΩ) was measured with a 5-mV hyperpolarizing pulse in voltage-clamp and left uncompensated. Membrane

potentials were corrected for a ~7 mV liquid junction potential. In some cases after the recording was complete, cellular morphology was captured in a volume stack using 740 nm two-photon laser light (Coherent). To activate ChR2 in acute slices from *ChAT* [i-Cre]; *Rosa26* [lsl-ChR2-EYFP] mice, 473 nm laser light (Optoengine) was focused onto the back aperture of the 60× water immersion objective to produce collimated whole-field illumination. Square pulses of laser light were delivered every 20 s and power (2–7 ms; 4.4 mW·mm$^{-2}$) was quantified for each stimulation by measuring light diverted to a focal plane calibrated photodiode through a low-pass dichroic filter. Following bath application of TTX and 4AP, in some cases, light power or duration was increased slightly to recover currents (e.g., changing the duration from 2 to 4 ms).

## Reagents

Drugs (all from Tocris, United Kingdom) were applied via bath perfusion: SR95531 (10 µM), tetrodotoxin (TTX; 1 µM), 4-aminopyridine (4AP; 500 µM), scopolamine (10 µM), 2,3-dihydroxy-6-nitro-7-sulfamoyl-benzo(*f*)quinoxaline (NBQX; 10 µM), *R*,*S*-3-(2-carboxypiperazin-4-yl)propyl-1-phosphonic acid (CPP; 10 µM), *N*,2,3,3-Tetramethylbicyclo[2.2.1]heptan-2-amine, (MEC; 10 µM), [1α,4(*S*),6β,14α,16β]-20-Ethyl-1,6,14,16-tetramethoxy-4-[[[2-(3-methyl-2,5-dioxo-1-pyrrolidinyl)benzoyl]oxy]methyl]aconitane-7,8-diol (MLA; 0.1 µM), (2*S*,13b*S*)-2-Methoxy-2,3,5,6,8,9,10,13-octahydro-1*H*,12*H*-benzo[*i*]pyrano[3,4-*g*]indolizin-12-one (DHβE; 10 µM). CPP and NBQX were combined to make a cocktail of antagonists to target ionotropic glutamate receptors, while MEC, MLA, and DHβE were combined to make a cocktail to antagonize nicotinic receptors.

## Acute slice data acquisition and analysis

Membrane currents and potentials were recorded using an Axoclamp 700B amplifier (Molecular Devices, Sunnyvale, CA) filtered at 3 kHz and digitized at 10 kHz using National Instruments acquisition boards and ScanImage (available at: scanimage.org) written in MATLAB (Mathworks, Natick, MA). Electrophysiology and imaging data were analyzed offline using Igor Pro (Wavemetrics, Lake Oswego, OR), ImageJ (NIH, Bethesda, MD) and GraphPad Prism (GraphPad Software, La Jolla, CA). In figures, voltage-clamp traces represent the average waveform of 3–6 acquisitions. Peak current amplitudes were calculated by averaging over a 1 ms window around the peak. For pharmacological analyses, 3–7 consecutive acquisitions (20 s inter-stimulus interval) were averaged following a 3-min wash-in period for NBQX and CPP or a 4-min wash-in period for MEC, MLA, and DHβE. For TTX and 4AP conditions, current averages were composed of the acquisitions following full block or first-recovery of ChR2 evoked currents, respectively. Data (reported in text and figures as mean ± sem) were compared statistically using the Mann–Whitney test. p values smaller than 0.05 were considered statistically significant.

## Acknowledgements

The authors thank members of the Sabatini Lab for helpful discussions related to this project and manuscript. We thank the Neurobiology Department and the Neurobiology Imaging Facility for consultation and instrument availability that supported this work. This facility is supported in part by the Neural Imaging Center as part of an NINDS P30 Core Center grant #NS072030.

## Additional information

### Funding

| Funder | Grant reference |
| --- | --- |
| National Institute of Neurological Disorders and Stroke (NINDS) | #NS072030 |

The funder had no role in study design, data collection and interpretation, or the decision to submit the work for publication.

### Author contributions

AS, Conception and design, Acquisition of data, Analysis and interpretation of data, Drafting or revising the article; AJG, Acquisition of data, Analysis and interpretation of data, Drafting or revising

the article; BLS, Conception and design, Analysis and interpretation of data, Drafting or revising the article

## Ethics

Animal experimentation: All experimental manipulations were performed in accordance with a protocol (#03551) approved by the Harvard Standing Committee on Animal Care following guidelines described in the US National Institutes of Health Guide for the Care and Use of Laboratory Animals.

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
