## [Decision Letter]

Thank you for sending your work entitled “Corelease of acetylcholine and GABA from cholinergic forebrain neurons” for consideration at *eLife*. Your article has been favorably evaluated by a Senior editor and three reviewers, one of whom, Sacha Nelson, is a member of our Board of Reviewing Editors.

The Reviewing editor and the other reviewers discussed their comments before we reached this decision, and the Reviewing editor has assembled the following comments to help you prepare a revised submission.

Because the reviews are concise and raise only minor concerns the Reviewing editor has included them in full here.

*Reviewer 1*:

This is a short straightforward report that uses mouse genetics, anatomical colocalization and physiology and pharmacology in brain slices to demonstrate that most forebrain cholinergic neurons are also GABAergic. Although there was one prior report hinting at this relationship based on marker expression, the results are largely surprising and change the way that we think about the function of cholinergic input to the cortex and other forebrain regions.

I have no substantive concerns except that it would be important to insure that there is not significant overlap with the other paper from this group currently in press.

There are two more minor suggestions for improvement:

1) It is slightly surprising that only 9/58 cells exhibited direct IPSCs (vs. 41/58 cholinergic EPSCs) while the overwhelming majority of cholinergic neurons examined expressed *Vgat* and *Gad2*. What if anything additional do the authors know about this? Did they try stimulating repeatedly to see if multiple action potentials may be required to release GABA at some synapses? Do they think there may be small IPSCs to which they are insensitive? Is it feasible that some of the presynaptic neurons do not express *Vgat* and *Gad2* at the protein level? This latter point would be easy to test with immunohistochemistry.

If the authors have negative results that constrain the answers to these questions, describing them would be a useful addition. But at the very least they should speculate about the reason for the discrepancy.

2) What have the authors examined with respect to the identity of the L1 neurons that receive and don't receive GABAergic input? Are they indistinguishable in terms of firing properties, morphology etc.? Again, even negative information would be useful.

*Reviewer #2*:

The authors examine the putative corelease properties of cholinergic synapses in the cortex using a combination of transgenic and optogenetic approaches. Post-synaptic currents were recorded in layer 1 interneurons following the selective activation of ChAT^+^ inputs in acute slice. The authors present several lines of evidence which support the direct release of GABA from cholinergic terminals, including: 1) light-induced, nAChR-antagonist resistant IPSCs, 2) expression of the enzymatic machinery for GABA biosynthesis and packaging, and 3) elimination of putative monosynaptic IPSCs in conditional *Vgat* knockout mice. These findings may lead us to re-evaluate the role of the basal forebrain cholinergic system in modulating cortical excitability and thus, are of interest to a broad neuroscience community. The specific functional implications of ACh/GABA cotransmission in the cortex are somewhat unclear and warrant further study.

Minor comments:

1) It appears that nAChR-resistant IPSCs are observed only in a subpopulation (approximately 15%) of recorded cells (as described in the Results section). However, IHC data suggest that molecular machinery for the synthesis and packaging of GABA are co-expressed in nearly all cortical ChAT^+^ cells. How might this discrepancy be explained? Since the direct release of GABA occurs at few cholinergic synapses, can we expect that this phenomenon will have significant impact on cortical function?

2) Can the quality of images in Figures 2 and 3 be improved, perhaps by higher magnification photomicrographs, to more clearly represent the quantification in panels B?

3) The current onset data (see panel H in Figure 1 for reference) should be included in Figure 4 to demonstrate the specific loss of direct, monosynaptic IPSCs in *Vgat*-deficient mice?

4) Were ACh-mediated EPSCs prolonged in *Vgat*-deficient mice? It appears so based on the trace shown in Figure 4. What would be the explanation of this result?

*Reviewer #3*:

The authors report the corelease of GABA and ACh onto cortical layer 1 interneurons. This is a meticulous study employing several transgenic mice, immunohistochemistry, pharmacology, optogenics and electrophysiology. The resulting interpretation of corelease is convincing.

Minor comments:

The only minor question is whether individual neurons release both transmitters. This seems highly likely as all presynaptic neurons, i.e., those expressing ChR2, are Chat positive and many/most are also positive for *Gad2* and *Vgat* (though why only ∼20% of the postsynaptic cells that have AChergic EPSCs also have GABAergic IPSCs is unclear; maybe the cited Nature paper explains this). Formally, however, an absolute demonstration of corelease would require the excitation of single presynaptic neuron or finding that minis have both components. This is a minor comment and certainly not necessary for this paper.

Also, the stated rationale for performing the experiment in Figure 4 (“the possibility remains that the direct IPSCs […] were due to nicotinic axon-axonic synapses onto GABAergic terminals”) seems to be incorrect. While multiple experiments leading to the same conclusion is always more satisfying that single experiments, it would seem that the nicotinic blocker experiment would also rule out the axo-axonic connections. Maybe a more reasonable introduction to Figure 4 would be something like: “Consistent with the pharmacology…” That said, the experiments in Figure 4 are elegant and convincing and should be included.

Do the authors have to use “impact” as a verb (in the Discussion)? Whatever happened to “affect”?

---

## [Author Response]

*Because the reviews are concise and raise only minor concerns the Reviewing editor has included them in full here*.

Reviewer 1:

*This is a short straightforward report that uses mouse genetics, anatomical colocalization and physiology and pharmacology in brain slices to demonstrate that most forebrain cholinergic neurons are also GABAergic. Although there was one prior report hinting at this relationship based on marker expression, the results are largely surprising and change the way that we think about the function of cholinergic input to the cortex and other forebrain regions*.

*I have no substantive concerns except that it would be important to insure that there is not significant overlap with the other paper from this group currently in press*.

We thank the reviewer for their support of our study. We too were surprised by the GABA corelease from cholinergic neurons and agree that this phenomenon will force a reconsideration of cholinergic modulation of forebrain structures.

Our paper in press at Nature is a detailed anatomical and functional description of a subcortical projection system to cortex. We discovered that the globus pallidus externus (GPe) sends a direct projection to cortex, thus altering the classical model of the basal ganglia. This projection is carried by two cell types, one of which is ChAT^+^ neurons of the GPe, which we demonstrate express *Gad2* and *Vgat* and release GABA in addition to ACh. That finding is a relatively minor part of the paper. Furthermore, as referenced by the referee above, *Gad2 and Vgat* coexpression was known to occur in cells of that region.

Corelease of GABA and ACh from the GPe ChAT^+^ cells inspired us to look at the potential of corelease in other cholinergic centers throughout the brain, resulting in the study submitted to *eLife*. The results presented here do not significantly overlap with those of the in press study. Furthermore the *eLife* study includes in depth analysis such as the conditional knock out of *Vgat* from Chat^+^ neurons, which provides genetic evidence supporting that GABA is released directly from cholinergic neurons.

*There are two more minor suggestions for improvement*:

*1) It is slightly surprising that only 9/58 cells exhibited direct IPSCs (vs. 41/58 cholinergic EPSCs) while the overwhelming majority of cholinergic neurons examined expressed* Vgat *and* Gad2*. What if anything additional do the authors know about this? Did they try stimulating repeatedly to see if multiple action potentials may be required to release GABA at some synapses? Do they think there may be small IPSCs to which they are insensitive? Is it feasible that some of the presynaptic neurons do not express* Vgat *and* Gad2 *at the protein level? This latter point would be easy to test with immunohistochemistry*.

*If the authors have negative results that constrain the answers to these questions, describing them would be a useful addition. But at the very least they should speculate about the reason for the discrepancy*.

We thank the reviewer for raising this interesting point regarding differences in the occurrence of GABA and ACh driven synaptic currents. We have expanded the discussion of this point in the manuscript.

In a subset of experiments, we tried stronger single-pulse or pulse-trains of ChR2 activation, but these alternative stimulations did not reveal GABAergic currents. Of course we cannot rule out the possibility that our somatic recordings were not able to resolve small IPSCs or heavily filtered IPSCs resulting from the activation of synapses on distal dendrites.

Our guess is that there is high specificity of the cortical targets of GABA release from Chat^+^ neurons. Several possibilities could explain target specificity of GABA/ACh currents. If GABA and ACh are co-packaged the same vesicles, the recorded post-synaptic neuron could express either GABA_A_R or nAChR. Alternatively, differential GABA/ACh corelease could be achieved presynaptically either by: 1) different types of neurons within the cholinergic system (for example local ChAT^+^ interneuron vs. subcortical projections), or 2) through independent release sites from single axons. Precedence in the literature exists for the latter possibility, as ACh and GABA may be independently released from starburst amacrine cells of the retina ([13]. Neuron 68: 1159-72. doi: 10.1016/j.neuron.2010.11.031).

There could be additional regulation of release by activity, neuromodulators or behavioral state, including post-transcriptional regulation of *Vgat* and *Gad2* mRNAs. Establishing the mechanisms of corelease is a priority of ongoing experiments in the lab.

*2) What have the authors examined with respect to the identity of the L1 neurons that receive and don't receive GABAergic input? Are they indistinguishable in terms of firing properties, morphology etc.? Again, even negative information would be useful*.

We have not examined the identity of L1 neurons that receive or do not receive GABA currents in detail. In many of our recordings, we capture the morphology of the neurons with 2P stacks. At the level of dendritic anatomy, there are not any striking differences, though we have not attempted a quantitative comparison of morphological properties. Biocytin-based morphological reconstructions would be necessary to test for differences in axonal anatomy, like the distinct flavors of L1 neurons that innervate the same or different cortical columns (Jiang et al. 2012. Nat Neuro. doi: 10.1038/nn.3305).

Our goal was to use the L1 neurons as a read out for both GABA and ACh mediated post-synaptic currents. Therefore we performed all of our analyses in voltage clamp with a Cs-based, low Cl^-^ internal solution, discriminating currents by holding potential and pharmacology. We felt that these recording conditions offered the highest sensitivity, as necessary to detect potentially small EPSCs/IPSCs. However, these recording conditions precluded us from looking at differences in active membrane properties. We are now embarking on a large effort to discover the rules for selectivity of GABA and ACh neurotransmission using a large panel of GFP marker lines and post-hoc analysis. We expect this effort will take a long time to complete.

Reviewer #2:

*The authors examine the putative corelease properties of cholinergic synapses in the cortex using a combination of transgenic and optogenetic approaches. Post-synaptic currents were recorded in layer 1 interneurons following the selective activation of ChAT*^*+*^
*inputs in acute slice. The authors present several lines of evidence which support the direct release of GABA from cholinergic terminals, including: 1) light-induced, nAChR-antagonist resistant IPSCs, 2) expression of the enzymatic machinery for GABA biosynthesis and packaging, and 3) elimination of putative monosynaptic IPSCs in conditional* Vgat *knockout mice. These findings may lead us to re-evaluate the role of the basal forebrain cholinergic system in modulating cortical excitability and thus, are of interest to a broad neuroscience community. The specific functional implications of ACh/GABA cotransmission in the cortex are somewhat unclear and warrant further study*.

We thank the Reviewer for their comments. We agree that GABA corelease from ChAT^+^ cells of the forebrain forces an important reconsideration for how the cholinergic system regulates excitability and activity in neural circuits. We also agree that the study presented here is an initial description of the ACh/GABA corelease phenomenon; future work needs to be done to understand the functional impact of GABA release throughout the many cortical and subcortical cell types and networks which receive ACh/GABA synapses.

*Minor comments*:

*1) It appears that nAChR-resistant IPSCs are observed only in a subpopulation (approximately 15%) of recorded cells (as described in the Results section). However, IHC data suggest that molecular machinery for the synthesis and packaging of GABA are co-expressed in nearly all cortical ChAT*^*+*^
*cells. How might this discrepancy be explained? Since the direct release of GABA occurs at few cholinergic synapses, can we expect that this phenomenon will have significant impact on cortical function*?

Please see reviewer #1, comment 1, in which we speculate on several corelease mechanisms that could provide target specificity of GABA and Ach release.

The impact of corelease on cortical function is an active area of research in the lab, necessitating acute slice circuitry/anatomy work, in vivo optogenetics and behavioral work in the context of conditional mouse genetics to isolate GABA vs. ACh signaling in the cholinergic system. Furthermore, it requires careful comparison of the effects on cortical function of activity in cholinergic systems with and without GABA release. This is, of course, a major research endeavor.

*2) Can the quality of images in*
Figures 2 and 3
*be improved, perhaps by higher magnification photomicrographs, to more clearly represent the quantification in panels B*?

We have now included high magnification insets for a clearer representation of the co-localization analysis. We left the low magnification images intact to illustrate the labeling patterns unique to each brain region and to illustrate the co-localization across multiple cells.

We have also added a figure supplement, showing that outside of the cortex, ChAT^+^ neurons do not coexpress GAD67, as described in our original submission.

*3) The current onset data (see panel H in*
Figure 1
*for reference) should be included in*
Figure 4
*to demonstrate the specific loss of direct, monosynaptic IPSCs in* Vgat*-deficient mice*?

The reviewer makes a good point. However, in our optogenetics experiments with the conditional *Vgat* allele, we focused on screening for direct GABA/ACh currents. Direct, monosynaptic currents were reliably evoked and quantified from ∼ 10 acquisitions. However the small number of acquisitions used for the conditional *Vgat* experiments were not conducive to detecting and quantifying indirect IPSCs, which are more unreliable and have less consistent onset times.

Nonetheless, we prepared a direct current onset latency figure for the reviewers (Figure 5). Since we did not detect any direct GABAergic IPSCs in the *Vgat*^fl/fl^ condition, there is no latency data. For the *Vgat*^+/+^ condition, GABAergic IPSCs were reliable and fast, as in Figure 1. Direct nEPSCs did not significantly differ in their mean (±sem) onset latencies across *Vgat* conditions (*Vgat*^fl/fl^: 5.1±0.5, n=12 cells; *Vgat*^+/+^: 4.2±0.7, n=7).Author response image 1.

*4) Were ACh-mediated EPSCs prolonged in Vgat-deficient mice? It appears so based on the trace shown in*
Figure 4*. What would be the explanation of this result*?

The kinetics of nEPSCs can vary widely due to different contributions of fast and slow components. The fast component is thought to be due to nAChRs containing α7-subunits and activated by point-to-point synaptic transmission. The slow component is thought to be due to non-α7 subunit containing nAChRs and volume transmission. In L1 interneurons, which have served as a model system for working out some of the properties of ionotropic ACh transmission, nEPSCs are quite variable ([2]. J Neurosci. doi: 10.1523/JNEUROSCI.3565-12.2012; [1]. J Neurosci. doi: 10.1523/JNEUROSCI.0115-12.2012).

Based on the reviewers suggestion, we rechecked the kinetics of nEPSCs across the floxed *Vgat* allele conditions. nEPSCs also had highly variable fast and slow components in both conditional KO and wild-type mice. The slightly prolonged fast component of *Vgat*^fl/fl^ is present in the normalized average nEPSC waveform (shown with sem, Figure 5). However, given the diversity in kinetics in both conditions, we are not able to determine if this difference is systematic or due sampling bias from small numbers. Many more recordings would be required to define systematic changes in nEPSC kinetics.

Reviewer #3:

*The authors report the corelease of GABA and ACh onto cortical layer 1 interneurons. This is a meticulous study employing several transgenic mice, immunohistochemistry, pharmacology, optogenics and electrophysiology. The resulting interpretation of corelease is convincing*.

We thank the reviewer for describing our study as “meticulous” and “convincing.”

*Minor comments*:

*The only minor question is whether individual neurons release both transmitters. This seems highly likely as all presynaptic neurons, i.e., those expressing ChR2, are Chat positive and many/most are also positive for* Gad2 *and* Vgat *(though why only ∼20% of the postsynaptic cells that have AChergic EPSCs also have GABAergic IPSCs is unclear; maybe the cited Nature paper explains this). Formally, however, an absolute demonstration of corelease would require the excitation of single presynaptic neuron or finding that minis have both components. This is a minor comment and certainly not necessary for this paper*.

Reviewer 3 makes an important point regarding the formal definition of corelease. We also appreciate the sentiment that resolving the mechanisms of corelease for multiple cell types within the cholinergic system should be reserved for future work. In regards to the corelease mechanisms, please see our response to referee #1, comment 1.

*Also, the stated rationale for performing the experiment in*
Figure 4
*(*“*the possibility remains that the direct IPSCs […] were due to nicotinic axon-axonic synapses onto GABAergic terminals*”*) seems to be incorrect. While multiple experiments leading to the same conclusion is always more satisfying that single experiments, it would seem that the nicotinic blocker experiment would also rule out the axo-axonic connections. Maybe a more reasonable introduction to*
Figure 4
*would be something like:* “*Consistent with the pharmacology…*” *That said, the experiments in*
Figure 4
*are elegant and convincing and should be included*.

We thank the reviewer for pointing out this miscommunication of rationale. Our intention was to suggest that because of the known complexities in receptor makeup of brain nAChRs, and the resulting complexities in antagonist pharmacology, that our blocker experiments could be prone to cryptic, resistant nAChRs. Nevertheless, we agree with the reviewer and have changed the wording accordingly.

*Do the authors have to use* “*impact*” *as a verb (in the Discussion)? Whatever happened to* “*affect*”?

“Affect” is making a resurgence! We have changed the wording as per the reviewer’s suggestion.